# Short and Long Term Clinical and Immunologic Follow up after Bone Marrow Mesenchymal Stromal Cell Therapy in Progressive Multiple Sclerosis—A Phase I Study

**DOI:** 10.3390/jcm8122102

**Published:** 2019-12-02

**Authors:** Ellen Iacobaeus, Nadir Kadri, Katia Lefsihane, Erik Boberg, Caroline Gavin, Anton Törnqvist Andrén, Anders Lilja, Lou Brundin, Katarina Le Blanc

**Affiliations:** 1Department of Laboratory Medicine, Division of Clinical Immunology, Karolinska Institute, 141 52 Stockholm, Sweden; nadir.kadri@ki.se (N.K.); katia.lefsihane@ki.se (K.L.); Erik.Boberg@ki.se (E.B.); Caroline.Gavin@ki.se (C.G.); anton.tornqvist.andren@ki.se (A.T.A.); Katarina.Leblanc@ki.se (K.L.B.); 2Department of Clinical Neuroscience, Division of Neurology, Karolinska Institute, 171 64 Stockholm, Sweden; Lou.Brundin@ki.se; 3Department of Clinical Neuroscience, Section of Neuroradiology, Karolinska Institute, 171 64 Stockholm, Sweden; Anders.J.Lilja@sll.se; 4Center for Haematology and CAST, Karolinska University Hospital Huddinge, 141 57 Stockholm, Sweden

**Keywords:** clinical trial, cell- and tissue-based therapy, mesenchymal stromal cells, multiple sclerosis, chronic progressive, autoimmune disease

## Abstract

Bone marrow derived mesenchymal stromal cells (BM-MSCs) have emerged as a possible new therapy for Multiple Sclerosis (MS), however studies regarding efficacy and in vivo immune response have been limited and inconclusive. We conducted a phase I clinical study assessing safety and clinical and peripheral immune responses after MSC therapy in MS. Seven patients with progressive MS were intravenously infused with a single dose of autologous MSC (1–2 × 10^6^ MSCs/kg body weight). The infusions were safe and well tolerated when given during clinical remission. Five out of seven patients completed the follow up of 48 weeks post-infusion. Brain magnetic resonance imaging (MRI) showed the absence of new T2 lesions at 12 weeks in 5/6 patients, while 3/5 had accumulated new T2 lesions at 48 weeks. Patient expanded disability status scales (EDSS) were stable in 6/6 at 12 weeks but declined in 3/5 patients at 48 weeks. Early changes of circulating microRNA levels (2 h) and increased proportion of FOXP3^+^ Tregs were detected at 7 days post-infusion compared to baseline levels. In conclusion, MSC therapy was safe and well tolerated and is associated with possible transient beneficial clinical and peripheral immunotolerogenic effects.

## 1. Introduction

Multiple Sclerosis (MS) is a chronic neuroinflammatory disease that results in considerable neurological disabilities. Most patients experience an early relapsing remitting MS (RRMS) disease phase, followed by a secondary progressive disease course (SPMS), which is characterized by continuous worsening of the clinical condition [1]. A minority of patients (15%–20%) display a clinical progressive disease from the onset (primary progressive MS; PPMS) which, as seen in SPMS, results in a gradual accumulation of disabilities over time [2]. 

Recent years have seen encouraging progress regarding treatment options for MS, with new highly effective MS disease modifying therapies (DMTs) launched [3]. Therapeutic responses to these new, highly effective DMTs are mainly associated with T- and B-cell immunosuppressive effects [3]. However, none of the current MS treatments have shown convincing effects on the neurodegenerative features of MS pathology [4]. This strongly limits their efficacy in the treatment of progressive MS [5]. In addition, treatment failure is reported in patients with aggressive MS and importantly, all of these new DMTs have associated potential severe side effects such as opportunistic infections and triggering of secondary autoimmune diseases [6]. There is, therefore, a call for the development of novel therapies targeting both the inflammatory and neurodegenerative components of MS disease pathology.

Mesenchymal stromal cells (MSCs) hold great promise as a cellular therapy, receiving approval as a standard line of care in countries such as Canada and New Zealand for certain clinical indications [7]. The mechanisms by which MSCs exert their therapeutic effects remain only partially elucidated but appears to be specific to the cell source and their mode of administration to the patient [8]. MSCs were originally isolated from the bone marrow (BM), but later also from several adult and fetal human tissues, with the observation of heterogeneous phenotypic and functional properties depending on the source of origin [7,9]. As highlighted in the current editorial review by Najar et al., the differences in immunological and regenerative profiles between tissue sources are associated with potential therapeutic and safety variations which have challenged the development of MSC based therapies [9]. Within the context of intravenous (IV) delivery, BM MSCs have been shown to trigger an instant blood mediated inflammatory reaction (IBMIR), which contributes to the downstream regulation of both innate and adaptive immune responses [10,11,12]. Patient specific responses, due to the role of the extracellular environmental milieu, are also thought to play a central role in MSC mode of action, demonstrating the need for further translational and mechanistic studies to improve efficacy and in predicting responsiveness [13,14,15,16]. 

The ability of MSCs to induce immune tolerance and skew responses away from the pathogenic has seen increasing research regarding their translation in the treatment of autoimmune diseases, including MS [17,18,19,20,21]. Their potential benefit in the treatment of MS has been highlighted through the use of acute and chronic models of experimental autoimmune encephalomyelitis (EAE) [22,23,24,25]. A milder EAE disease course following MSC transplantation has been associated with reduced axonal loss and demyelination, with associated neuroprotective effects explained by antioxidative and anti-apoptotic mechanisms [26,27]. Furthermore, therapeutic effects of IV infused MSCs in these models have been associated with systemic immune dampening, such as the induction of T cell anergy and reduced production of pro-inflammatory cytokines [22,24]. 

Although insights into biological mechanisms of action that are associated with a favorable in vivo response of MSC therapy have been delineated in EAE, similar studies are limited in the human setting [22,23,24,25,26,28,29,30]. Most human studies have focused on safety and clinical outcome measures. Major conclusions about clinical benefit in MS have, however, been difficult to draw due to small, heterogenous study cohorts, in addition to differences in MSC source and administrative route [17,31,32,33,34,35,36]. A possible therapeutic effect of IV and intrathecal (IT) MSC infusion, supported by a decrease in mean expanded disability status scale (EDSS) [37], was reported in one study on treatment refractory MS [17]. Evidence of tissue repair, shown by the improvement of visual acuity and increased optic nerve area, was detected in SPMS six months after IV MSC therapy [33]. Furthermore, a possible reduction in gadolinium enhancing (Gd+) brain magnetic resonance imaging (MRI) lesions was reported in a small randomized phase II cross-over study assessing IV MSC therapy in RRMS [35]. These results were, however, in contrast to those reported by Cohen et al. who found no clear evidence of inhibition of disease activity or tissue repair in RRMS and SPMS [36]. 

Whilst differences in suggested efficacy and therapeutic benefit have been reported with MSC administration, one notable observation in clinical trials specifically using BM MSC across several diseases has been the high safety profile, tolerability, and lack of severe adverse events (AE) reported since the very earliest studies [17,32,33,34,35,36,38]. This notion underscores the validity and value in further investigating the use of BM MSC therapy as a possible new treatment option for MS.

Comprehensive immunological profiling at a cellular and molecular level post-MSC infusion in MS has been poorly explored but will be of priority in delineating MSC mode of action and the role of the patient’s immune repertoire in predicting responsiveness to therapy and to improve clinical efficacy. Experience from clinical trials on MSC therapy in other autoimmune diseases including diabetes mellitus (DM), rheumatoid arthritis (RA), Sjögren’s Syndrome (SS), and Systemic Lupus Erythematosus (SLE) has suggested that systemic tolerance and induction of regulatory T cells (Tregs) following IV infusion were associated with beneficial therapeutic effects [19,20,21,39].

The clinical experience of BM MSC therapy in MS is mainly based on studies using autologous cells. However, a few reports have recently demonstrated inferior in vitro immunosuppressive and neuroprotective properties of BM MSCs derived from MS compared to healthy donors (HD) [40,41]. Such findings may be attributed to chronic inflammation and a possible dysfunctional regulation of the BM niche, which raises the concern whether autologous cells may have disadvantages compared to their allogeneic counterpart. 

Here, we report findings from our phase I study in MS patients to assess the safety and clinical response to a single-dose IV infusion of autologous BM MSCs, with paralleled peripheral immune profiling pre- and post-infusion. Furthermore, in vitro phenotypic and immunosuppressive analyses of the clinical graded MSCs that were used in the current trial were compared with MSCs of HD. 

## 2. Material and Methods

### 2.1. Clinical Study Design

This was an open-label phase I, single-center, pre- and post-comparison study on the safety and clinical response to autologous MSC therapy in MS. The primary objective was the evaluation of safety, recorded as AE. The co-primary objective was the impact on MS disease activity as measured by the number of new MRI T2 lesions (defined as high signal lesions identified on 3D T2 FLAIR images). MRI protocol is provided in Appendix A (See Appendix A). The secondary objective was clinical outcome, which was assessed by the change in EDSS and number of relapses, with a co-secondary objective establishing alterations in peripheral immune cell populations. The patients were followed for 48 weeks post MSC therapy and the study was registered within ClinicalTrials.gov, identifier: NCT03778333.

### 2.2. MS Subjects, Healthy MSC Donors, and Ethics

Eligible patients had a diagnosis of MS according to the 2010 McDonald criteria [42]; a disease course corresponding to RRMS, SPMS, or PPMS; were aged between 18–50 years; and had baseline EDSS 3.0–7.0 and MS disease duration of 2–20 years, with signs of clinical deterioration of MS in the previous 1 to 2 years and failure to respond to treatment with DMTs. Patients that had been treated with any immunosuppressive therapy within 3 months prior to enrollment or with interferon-beta, glatiramer acetate, or corticosteroids within 30 days prior were excluded. Further inclusion and exclusion criteria are provided in Appendix A. The patients were enrolled between October 2012 and January 2015. Regional ethics approval (Dnr 2011/1810-31/3, 2014/822-31/4 and 446/00) was obtained and the study was conducted in accordance with the principles of the Declaration of Helsinki with informed written consent obtained from the MS subjects and HD.

### 2.3. MSC Therapy

Posterior iliac crest BM aspiration was performed (80–100 mL aspirate) under local anesthesia at the Department of Hematology, Karolinska University Hospital, Huddinge, Sweden. Clinical MSC production was performed at Vecura, Karolinska Institute, Huddinge, Sweden under good clinical practice conditions according to the guidelines of the European Blood and Marrow Transplantation Group [43]. At the time of the study, MSC was classified as a tissue and production was approved by the Swedish National Board of Health and Welfare. Briefly, after density gradient separation, BM mononuclear cells were seeded at a density of 160,000 cells/cm^2^ in Dulbecco’s Modified Eagle’s Low Glucose Medium (DMEM-LG, Life Technologies, Gaithersburg, MD, USA) supplemented with pooled platelet lysate obtained from HD (final concentration 10^8^ platelets/mL; purchased from Karolinska University Hospital, Huddinge, Sweden). Cells were detached using 0.05% (v/v) trypsin-EDTA (ThermoFisher Scientific Life Sciences, Heidelberg, Germany) when the cultures reached near confluence (>80%) and were re-plated at a density of 4000 cells/cm^2^. BM-MSCs were harvested at passage 1 to 2 and cryopreserved in 10% (v/v) dimethyl sulphoxide (WAK-Chemie Medical GmbH, Steinbach Germany) 3–6 weeks after BM aspiration. Flow cytometric analyses confirmed MSC expression of (>95%) CD73, CD90, CD105, human leukocyte antigen (HLA)-ABC and absence of (<5%) CD14, CD31, CD34, CD45, and HLA II cell surface expression. The release criteria for clinical MSCs were a viability >95%, absence of contamination by pathogens (bacteria and mycoplasma), and spindle shape morphology when attached to plastic. After thawing, the cells were washed and re-suspended in 0.9% (w/v) saline solution supplemented with 10% (v/v) AB plasma at a final concentration of 2 × 10^6^ cells/mL and were subsequently delivered to the patients without any culture recovery period. Each patient received a single IV infusion of 1–2 × 10^6^ cells/kg body weight. 

### 2.4. Safety

Vital signs (heart rate, blood pressure, percutaneous oxygen saturation) and AEs (recorded by history taking and physical examination) were monitored for 3 h following MSC infusion (15 min, 1 h, and 3 h). Further recording of AE and routine blood and urine analyses were assessed at baseline; 1, 3, and 7 days; and 4, 12, 24, 36, and 48 weeks post-infusion. 

### 2.5. Clinical Evaluation

The patients were evaluated with brain MRI (Appendix A) EDSS, the 9-Hole Peg Test (9-HPT) [44], the Symbol Digit Modalities Test (SDMT) [45], the Paced Auditory Serial Addition Test (PASAT) [44], the Multiple Sclerosis Impact Scale (MSIS-29) [46], the Fatigue Severity Scale (FSS) [47], and the health status scale EQ-5D [48] at baseline, 12, 24, 36, and 48 weeks post MSC infusion.

### 2.6. Preparation of Peripheral Blood Mononuclear Cells and Plasma 

Peripheral immune cell populations and plasma cytokine levels were analyzed at baseline; 2 h; 1, 3, and 7 days; and 4, 12, and 24 weeks post MSC treatment. Peripheral blood mononuclear cells (PBMCs) and plasma were obtained from blood samples collected in mononuclear cell preparation tubes (Vacutainer CPD: BD Biosciences, San Jose, CA, USA) by density gradient centrifugation as previously described [49]. Plasma was recovered by centrifugation at 1500× *g* for 15 min and samples were stored at −80 °C until use.

### 2.7. Flow Cytometric Analysis of Peripheral Immune Cells 

Frozen PBMCs were thawed and washed twice in RPMI 1640 medium (Life Technologies) supplemented with 10% (v/v) fetal calf serum (FCS; Life Technologies, New York, USA). Cell surface or intracellular staining (antibodies listed in Appendix A) was accomplished as described previously [49]. Cells were acquired using a FACS LSR-II Fortessa (Becton Dickinson and Company, San Jose, CA, USA) and data were analyzed using FlowJo Version 10.0 software (FlowJo, Ashland, OH, USA). The relative difference in immune cell frequency was normalized to baseline pre-MSC infusion levels, which was set at 100%. 

### 2.8. Cytokine Analyses and microRNA Analyses

Plasma cytokine levels of interleukin (IL) 1β, IL-13, IL-2, IL-17A, IL-17F, IL-6, IL-10, IL-12p70, IL-4, IL-23, IFN-γ, TNF-α, and vascular endothelial growth factor (VEGF) were evaluated using the LEGENDplex™ custom human 13-plex Panel (Biolegend, San Diego, USA) according to the manufacturer’s protocol. Fluorescence intensity for each analyte was detected using a FACS LSR-II Fortessa and data converted to concentrations using the provided standard curve. Changes in circulating microRNA (miR) levels in MS were assessed at 2 h, 24 h, and 3 days after MSC treatment. miR expression profiles in plasma were conducted by Exiqon Services, Qiagen, Denmark using the miRCURY LNA miRNA qPCR serum/plasma panel, incorporating assays for 179 miRs, which are most commonly found in the peripheral blood [50]. RNA spike-in tests were used to monitor RNA purification (UniSP2 and UniSP4) and cDNA synthesis (UniSP6). The analysis detected that all but one sample were suitable for profiling (one sample at +3 days post infusion), which was removed from the subsequent statistical analyses. KEGG analysis, for mapping of the biochemical pathways, was performed using DIANA-mirPath v3.0 software, (http://snf-515788.vm.okeanos.grnet.gr) (Appendix A) [51].

### 2.9. In Vitro Phenotypic and Immunosuppressive Analyses of MSC

MSC from HD (*n* = 5, 4 males, mean age 40 years; range 29–52 years) were isolated from BM aspirates as described above. In vitro analyses of MS MSC of patients that were included in the clinical trial (*n* = 6) and HD MSC were performed at passage 2 to 4 after expansion in DMEM-LG, supplemented with 10% (v/v) pooled platelet lysate and 1% amino acids, at 37 °C/5% CO^2^. Surface marker expression was analyzed using anti-CD45, anti-CD73, anti-CD34, anti-CD55, anti-CD59, anti-CD105, anti-CD90, anti-CD31, anti-HLA ABC, and anti-HLA DR antibodies (Biolegend) (antibodies are listed in S2). After 20 min of incubation at 4 °C, the cells were centrifuged for 5 min at 400× *g* at 4 °C. The supernatant was removed and the cell pellet of each well was taken up in 200 μL of PBS for subsequent flow cytometry analyses. For adipocyte differentiation, replacement to adipogenesis differentiation medium (Gibco) was performed. Cells were kept at 37°C/5% CO^2^ for 14 days, washed twice in PBS, and, thereafter, were fixed for 60 min at room temperature (RT) with 4% (w/v) paraformaldehyde (Sigma-Aldrich, Darmstadt, Germany). After the washing of the wells, 60% (v/v) isopropyl alcohol (Sigma-Aldrich) was added for 5 min, with the subsequent addition of 1% (w/v) Oil Red O reagent (dissolved in isopropyl alcohol) (Sigma-Aldrich) for 10 min, which was followed by washing 4 times with distilled water. The presence of lipid vacuoles were detected under a wide field optical microscope. For osteoblast differentiation, MSC were kept in osteoblast differentiation medium (Miltenyi Biotec, Auburn, USA) for 10 days. For visualization, cells were fixed as described above and stained with 2% (w/v) Alizarin Red (Sigma-Aldrich), diluted in distilled water, for 45 min at RT. Cells were then washed 4 times with distilled water. The presence of calcium deposits were detected under a wide field optical microscope. For T cell suppressive assays, PBMCs were isolated from HD buffy coats (*n* = 2) as described previously [49]. Briefly, CD3+ T cells (purity > 95%) were isolated using a Pan T cell negative selection kit (Miltenyi Biotec) following the manufacturer’s instructions. CD3+ cells were stained with 5 μmol L^−1^ carboxyfluorescein succinimidyl ester (CFSE; Invitrogen, Carlsbad, CA, USA) and activated using anti-CD3+ and anti-CD28+ microbeads (0.8 × 10^6^ /5 M CD3+ cells) (Miltenyi Biotec). Proliferative capacities in the presence or absence of MSC, at ratios of (MSC: CD3+) 1:0, 1:1, 1:2, 1:4, 1:8, 1:16, and 0:1, were assessed after five days of co-culture using flow cytometry (BD Fortessa LSR-II, NJ, USA). Data were analyzed using the FlowJo Version 10.0 software (Ashland, OH, USA). For MSC licensing, cells were stimulated in vitro with IFN-γ (100 IU/mL) and TNF-α in complete media (10 ng/mL) (RD systems) for three days. Cells were washed and stained with anti-CD73 antibody (Biolegend) and the live Aqua Cell Death marker (Invitrogen, Carlsbad, CA). After 20 min of incubation at 4 °C, MSC were centrifuged for 5 min at 400× *g* at 4 °C. Intracellular staining for indoleamine- 2,3-dioxygenase (IDO) and IL-6 (BD Biosciences) was performed. Data were analyzed using the FlowJo Version 10.0 software (FlowJo, Ashland, OH, USA).

### 2.10. Statistical Analyses

Clinical outcome measures and relative frequencies of peripheral immune cell populations were analyzed using the Wilcoxon signed-rank test, comparing data for each timepoint to the pre MSC infusion baseline. MiR levels were analyzed using a paired Student´s t-test. Low/undetectable plasma cytokine levels precluded any statistical analysis. Results from in vitro assessment of MSCs were analyzed using the Mann-Whitney test. P values were considered significant at the 0.05 level. Analyses were performed using R statistical software (R Foundation for Statistical Computing, Vienna, Austria) and Prism 5.0 (GraphPad, San Diego, CA, USA).

## 3. Results

### 3.1. Characteristics of Clinical Study Subjects

A schematic flow chart of the clinical study is shown in Figure 1. Seven (two PPMS, five SPMS), mean age 40 years, mean EDSS 6.5, and mean disease duration 12 years, out of ten screened patients were included in the study. Baseline characteristics are presented in Table 1. Prior DMTs included interferon-beta, glatiramer acetate, fingolimod, natalizumab, mitoxantrone, and rituximab. 

### 3.2. MSC Treatment and Safety

All study subjects received a single IV infusion of 1–2 × 10^6^ autologous MSCs/kg body weight, without pre-medication. Cells were obtained from passage 1 (*n* = 4), passage 2 (*n* = 1), or a combination from passage one and two (*n* = 2), and mean population doubling was 3.2 (range 2.5–4.6). No severe or serious AEs related to MSC infusion were reported. One patient developed a facial rash 6 h post-infusion, which resolved spontaneously over 24 h, and a second reported headaches during the first week after treatment. All AEs reported during follow-up are listed in Appendix A, with none possessing causative association with MSC therapy. Five patients completed the follow-up of 48 weeks during which they did not receive any additional immunomodulatory or immunosuppressive therapy. One patient discontinued study participation at 4 weeks post MSC treatment due to a severe relapse. The relapse had started 26 days prior to the MSC infusion, but neurological deterioration occurred after treatment. The patient declined from an EDSS of 3.5 at relapse onset to 5.5 at the point of MSC infusion, and 7.5 at 2 weeks post-infusion. Thereafter, she chose to discontinue study participation. Relapse symptoms included progressive paraparesis and sensory disturbance in the lower legs and trunk. The relapse did not respond to two cycles of IV steroids. The patient was consequently treated with 500 mg IV rituximab and underwent successful autologous hematopoietic stem cell transplantation (aHSCT) 3 months post-MSC infusion. The patient improved clinically and presented with an EDSS of 3.0 one year after MSC infusion. Another patient (patient 5) voluntarily discontinued the study at 36 weeks post-infusion after 2 relapses (week 24 and 36). Both relapses responded to IV steroids. The patient was treated with rituximab and presented with a stable EDSS, compared to baseline, at 48 weeks post-MSC infusion. All seven patients were followed for three years after the end of the study, as part of routine neurological care, without the observation of late AEs related to MSC therapy.

### 3.3. MRI and Clinical Assessment 

MRI and clinical outcome measures are shown in Figure 2. Baseline MRI, before MSC infusion, detected the absence of new lesions in all study subjects except in the patient that was infused during a relapse, who displayed 12 new T2 (including 10 Gd+) lesions. Patient 2 had one new T2 (Gd−) lesion at 12 weeks, patient 3 had one new T2 (Gd−) lesion at 48 weeks, patient 4 had one new T2 (Gd−) lesion at 24 weeks, and patient 5 had one new T2 (Gd−) and one T2 (Gd+) lesion at 24 weeks and two new T2 (Gd+) lesions at 36 weeks, concluding a total of seven new T2 lesions in 4/6 patients (of which 1/6 patients had Gd+ lesions) over 48 weeks (Figure 2A). EDSS remained stable in 6/6 patients at 12 weeks and in 5/6 at 24 weeks, however two patients had declined by 0.5 points and one patient had declined by 1 point at +48 weeks (Figure 2B). Hand function (9-HPT) and cognitive function (SDMT and PASAT) were stable during follow-up (Figure 2C–F). No obvious increasing or decreasing trends in patient reported outcomes (MSIS-29, EQ5D, and FSS) across the measures for each participant were detected during the follow-up of 48 weeks (Figure 2G–J).

### 3.4. Plasma Cytokine and miR Levels after MSC Infusion 

Longitudinal evaluation of all cytokines that were analyzed showed low or undetectable levels at all time points after MSC infusion in patient plasma. Plasma miR profiling was performed to assess possible short-term in vivo effects of MSC therapy. In total, an average of 167 out of the 179 miRs analyzed were detected. Significant changes were observed in 19 miRs 2 h after MSC infusion (Table 2). KEGG analysis of the overlap between these miRs indicated significant correlation with 66 pathways, of which those linked to the sphingolipid and mTOR signaling, proteoglycan regulation, including the transforming growth factor beta (TGFβ) signaling pathway, and T cell receptor signaling were of particular interest (Appendix A). Further analysis at 24 h post MSC infusion detected changes in one miRs (hsa-miR-332-3p) compared to pre-treatment levels. Analyses three days post treatment showed altered expression of four miRs (including hsa-miR-375, which was further increased compared to +2 h levels) (Table 2), of which hsa-miR-375, hsa-143-3p, and hsa-miR-29a-3p are significantly linked to 36 KEGG pathways including ECM-receptor interaction, steroid and fatty acid biosynthesis, and the FoxO signaling pathway. 

### 3.5. Alterations in the Peripheral Immune Repertoire after MSC Therapy

To investigate the in vivo effects of MSC infusion on peripheral immune response, longitudinal analysis of immune cell populations was performed by flow cytometry. Total leukocyte number and proportions of T (CD3+), B (CD19+), NK (CD56+), and NKT cells (CD56+CD3+) were similar at all time points before and after MSC treatment (Appendix A). We observed significantly increased CXCR3 expression on T, NK, and NKT cells at 2 h (*p* < 0.05) and 1 day post-infusion (*p* < 0.05), returning to baseline at 3 days post-infusion (Figure 3). It has been shown that MSC downregulate T cell proliferation by the induction of CXCL3 expression of T cells [52]. 

The differentiation status of CD4+ and CD8+ T cells was further assessed, demonstrating a decrease in CD4+CD27+CD45RA+ naïve cells (*p* < 0.05), in parallel with an increased level of CD4+CD27-CD45RA- effector memory cells (*p* < 0.05) at +4 weeks post infusion, compared to baseline levels (Figure 4). Similar changes in the CD8^+^ cell compartment were seen, with decreased levels of CD8^+^CD27^+^CD45RA^+^ naïve cells at +4 weeks and +12 weeks (*p* < 0.05). Increased levels of CD8+CD27-CD45RA- effector memory cells were detected at +4 weeks (*p* < 0.05) compared to baseline levels (Figure 4). Furthermore, a significant increase in the proportion of FOXP3^+^ Tregs at 7 days (*p* < 0.05) and 4 weeks (*p* < 0.05) post-infusion, compared to pre-treatment levels, was detected (Figure 5).

### 3.6. Comparable Characteristics of MSCs of MS Patients and Healthy Donors

Analyses of autologous MSCs derived from MS patients who were included in the clinical trial exhibited a number of similar properties compared to MSC of HD (Figure 6). The proliferative rate and multilineage potential, including adipocytic and osteoblastic differentiation, were comparable between MS and HD. No difference in MSC surface molecule expression, including those important for MSC survival after contact with blood (complement regulators), were detected [53]. Co-cultures of MSCs and CD3+ cells demonstrated T-cell suppressive capacities in a similar dose dependent manner in MS and HD. Intracellular expression of IDO and IL-6, after licensing, were comparable in MSC cultures of MS patients and HD. These data suggested that MSC phenotype and functions were not affected by the MS disease.

## 4. Discussion

The neuroprotective and immunoregulatory effects that were observed after MSC infusions in experimental models of MS have underpinned a rationale for clinical trials in MS patients. Specifically, subgroups of progressive MS are of interest due to the unmet therapeutic need for this cohort. In line with previous data, the current study demonstrated that autologous MSC therapy was safe and well tolerated in patients who were treated during clinical remission [31,32,33,34,35,36]. The early EDSS and MRI stability for up to 24 weeks in the majority of patients, after MSC therapy, may suggest a transient induction of clinical stability, considering that all patients displayed evidence of clinical deterioration within 1 to 2 years prior to study inclusion. Our findings are in agreement with previous studies reporting stable (or possible improvement) in EDSS at six months after MSC infusion [33,35,36]. 

Evidence that MSC infusion during a relapse (performed in one patient) failed to induce clinical improvement, but rather worsened neurological performance caught our attention. These findings highlight the role of the patient status at the point of infusion in responsiveness to MSC therapy. Paradoxical neuroinflammatory exacerbation after IT MSC infusion in a patient with severe myelitis has previously been reported [54], however similar reactivation in MS after IV MSC therapy, to the best of our knowledge, has not been described. Although the clinical deterioration cannot be clearly linked to MSC treatment, there is a need to critically consider inclusion/exclusion criteria in larger trials in order to delineate the efficacy of MSC treatments in MS. The current analyses of peripheral immune cell populations, cytokine, and miRNA levels prior to MSC infusion did not reveal any notable differences in the local mileu of this patient compared to the patients who were transfused during remission. However, treatment with IV steroids prior to MSC infusion may have dampened the peripheral inflammation. Furthermore, the in vitro assessment of the MSCs that were applied for clinical use demonstrated comparable phenotypic and immunosuppressive characteristics in this patient compared to the other patients and to HD. It is likely that the severe treatment refractory MS disease history in this patient rather explained the aggressive evolution of the relapse. 

The present detection of comparable in vitro phenotypic and immunosuppressive functions of MS MSC and HD MSC are in line with a few previous studies, while others reported on reduced immunosuppressive and neuroprotective function of MS MSC compared to HD MSC [40,41,55,56]. In light of the pronounced clinical effect of allogenic MSC therapy in EAE models, contrasting the modest effect of autologous MSC therapy shown in MS, it has been questioned whether MS MSC have reduced therapeutic capacities compared to HD MSC. The experience of allogenic MSC therapy in MS is very limited. A few reports have shown promising results of allogenic umbilical cord (UC) MSC therapy in MS [57]. Allogenic UC MSC may enable a more accessible and practical supply of cells for infusion compared to allogenic BM MSC. Additional factors with a possible impact on therapeutic potential are the route of administration and the in vitro preparation of MSCs. In the current study, freshly thawed cells were infused, which might have had an impact on the functionality of the cells. It has been shown that cryopreserved MSCs have decreased in vitro immunosuppressive properties compared to cryopreserved MSCs that were allowed to recover for 24 h post thawing [58]. However, the therapeutic and in vivo significance of in vitro phenotypic and functional signatures of MSCs remains to be further explored.

The presence of alterations in miR levels after MSC infusion are interesting since miR expression profiling in biological specimens of MS have provided evidence that dysregulated miRs play significant roles in immunopathogenic mechanisms in MS [59]. Circulating miRs are noncoding RNAs with diverse effects dependent on post-transcriptional regulation of gene expression. Secretion of miRs has been shown to be important for the therapeutic effect of MSCs and have been suggested to be a mechanism for communication between MSCs and other cells [60,61]. Correlative analysis confirmed that the miRs play a role in Hippo signaling, a pathway that has been previously linked to the pathogenesis of other autoimmune and neurological diseases including DM, amyotrophic lateral sclerosis, and Alzheimer’s disease [62]. Dysregulation of this pathway has also been linked to FoxO signaling, a second significantly linked KEGG pathway that was demonstrated in our miR analysis. 

The mechanism of action of IV MSC therapy has not been clearly elucidated, however previous studies have suggested that MSCs are promptly cleared from the circulation, possibly leading to their short-term accumulation in the lungs before clearance through the secondary lymphoid organs and other organs, with low long-term engraftment [11,63,64,65]. The detection of MSCs in the brain and spinal cord of EAE mice treated with MSC therapy have supported the hypothesis that the beneficial effects of IV MSC infusion are associated with their migration into the CNS [22,28]. This hypothesis was further corroborated by MRI studies that indicated the presence of labeled MSCs in the spinal cord of MS patients after combined IV and IT MSC infusion [17]. A possible therapeutic neuroprotective and immunomodulatory effect that is associated with CNS engraftment may consequently also be part of the mechanism of action, but remains to be further investigated. 

Our work, as well as the work of others, has highlighted the role of the complement cascade in triggering the IBMIR in response to MSC infusion [11,66]. Previous clinical findings have suggested that this may be a central mechanism of action by which MSCs are injured/killed, forcing the release of their intracellular contents and associated exosomes [11,67]. The findings from this study indicate changes in plasma borne miRs associated with pathways including the sphingolipid pathway; sphingolipid, with sphingosine-1-phosphate known to be released by MSCs; and other exposed cell types including platelets and endothelial cells in response to the coagulation cascade and complements [68]. 

One of the decreased miRs after MSC infusion was miR-155-5p, which has been reported as an important biomarker for MS disease activity [69]. Elevated levels of miR-155 have been detected in the blood, CSF, and brain resident cells of MS patients and EAE and in vitro studies have shown that overexpression of miR-155 promoted Th17 and Th1 pathways and enhanced neurodegenerative processes [69]. Furthermore, it was observed that miR-155 levels declined to normal levels after successful aHSCT in MS [70]. These observations may suggest, in the context of this study, that the decrease in miR-155-5p may be associated with a therapeutic beneficial effect mediated by IV MSC infusion. MiR pathway analysis also suggested changes in biochemical pathways linked to immunomodulation, such as the TGFß1 signaling pathway, T cell receptor, and FoxO signaling, indicating clear effects of MSC infusion within the first 2 h. These observations are further supported by current data demonstrating an early change in CXCR3 expression on CD3+ cells (2 and 24 h). Alterations in T cell subsets were not apparent until seven days post-infusion, strongly indicating that the MSCs induce a “hit and run” therapeutic effect, as demonstrated in other disease contexts [65]. Since antigen-presenting cells have a pivotal role in T cell regulation, in addition to substantial evidence that MSC mediated T cell modulation is dependent on monocytes/macrophages mechanisms, it may suggest that the T cell alterations occurred secondary to changes in these cells [71,72,73,74]. It has been shown that MSC inhibited myeloid cell functions by regulating chemokine, cytokine, and surface marker expression and had an impact on M1/M2 polarization, rendering the cells to obtain anti-inflammatory properties [72,75,76]. This finding is supported by the current observation of decreased miR-155-5p after MSC infusion, which has been reported to play a role in the mechanisms regulating M1/M2 profiling. In addition, a previous study showed significant reduction of CD86^+^, CD83^+^, and HLA^−^DR^+^ myeloid DCs at 24 h after MSC therapy in MS patients [17]. Assessment of peripheral monocyte activity after MSC infusion was not performed in the current study due to methodological challenges related to the usage of cryopreserved PBMCs.

The observed increase in CD4 and CD8 effector memory cells at four weeks after MSC infusion raises the question whether this population might contain unwanted autoreactive T cells [77]. An intuitive explanation of such an alteration may be that MSC infusion affects the migration of the T cell pool in different organs via their secretome [14]. This hypothesis is corroborated by our finding of an early change in CXCR3 surface expression of T cells. Interestingly, Muraro et al. showed that the proportion of effector memory CD4+ cells (CD4+CD45RA−CD45RO+CD27−) increased significantly at 6 months after successful aHSCT in MS, which argues against a pathogenic role for the observed induction of CD4 and CD8 effector memory cells [78]. Furthermore, increased levels of CD28+EM cells in natalizumab treated progressive MS was reported and interpreted as a result of beneficial retention of effector T cells in the systemic compartment [79]. Additionally, a nominally negative correlation between CD28-CD4+ EM T cells and MS disease progression was reported, arguing for a protective role of the EM T cell expansion in MS [79].

The current detection of a relative increase in the proportion of FOXP3 Tregs early after MSC infusion corroborates previous findings of increased relative levels of CD4+CD25+ Tregs at 4 and 24 h after combined IV and IT MSC infusion in MS [17]. Furthermore, our detection of restoration to baseline levels of Tregs at 24 weeks was in line with findings in RRMS treated with IV MSC, which presented with unchanged natural and induced Treg levels at 6 months after treatment [35]. Similarly, allogenic MSC therapy in SLE patients induced increased levels of CD4+FOXP3 cells among PBMCs at one week, three months, and six months after IV infusion, which correlated with amelioration in disease activity [19]. In addition, IV MSC infusion in patients with SS alleviated disease symptoms and directed T cells toward Tregs and Th2 cells [20]. Based on the prior substantial in vitro evidence that MSC induce Treg expansion by soluble factors, we hypothesize that similar mechanisms may be involved in the current observations. The observed early temporal increase in Tregs may suggest that multiple infusions of MSCs could induce prolonged therapeutic effects. The current MSC expansion up to passage 1 to 2 exceeded the total number of cells required for a single MSC infusion of 1–2 × 10^6^cells/kg in 6/7 patients. These cells could potentially have been further expanded, cryopreserved, and used for a second IV infusion. This suggestion is supported by a previous study, which shows the induction of long term clinical stability in SPMS after repeated IT infusions of MSCs [80]. 

## 5. Conclusions

In summary, we showed that IV MSC therapy in patients with stable progressive MS is safe and well tolerated. Immunoprofiling of patients suggests that MSCs trigger early effects, potentially as part of graft destruction via the complement cascade, leading to downstream secondary effects on the T cell repertoire. The detection of a transient increase in circulating Tregs may provide biologic rationale for repeated IV MSC infusions to induce long term tolerogenic effects and therapeutic significance. The encouraging results from this study exemplifies the need for data validation in a larger patient cohort in order to establish the efficacy of our MSC product. Our center is currently participating in the multicenter randomized double-blind cross-over clinical phase I/II study, “Mesenchymal Stem Cells for MS” (MESEMS) [81], which may provide more explicit conclusions on the clinical efficacy of BM MSC therapy in MS.

## Figures and Tables

**Figure 1 jcm-08-02102-f001:**
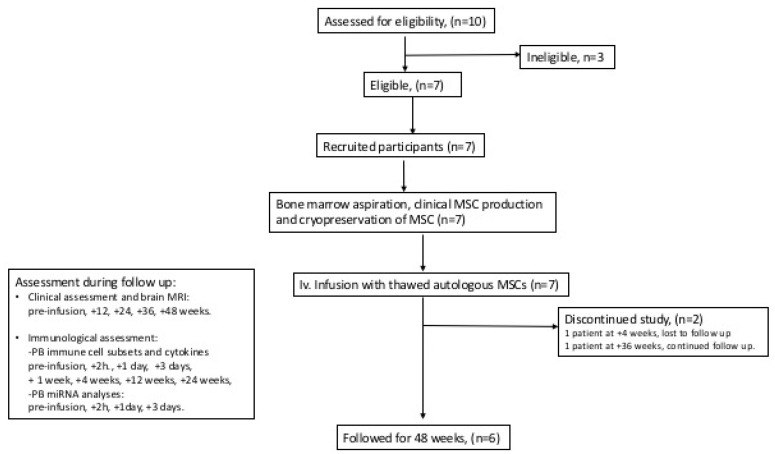
Schematic flow chart of the clinical trial.

**Figure 2 jcm-08-02102-f002:**
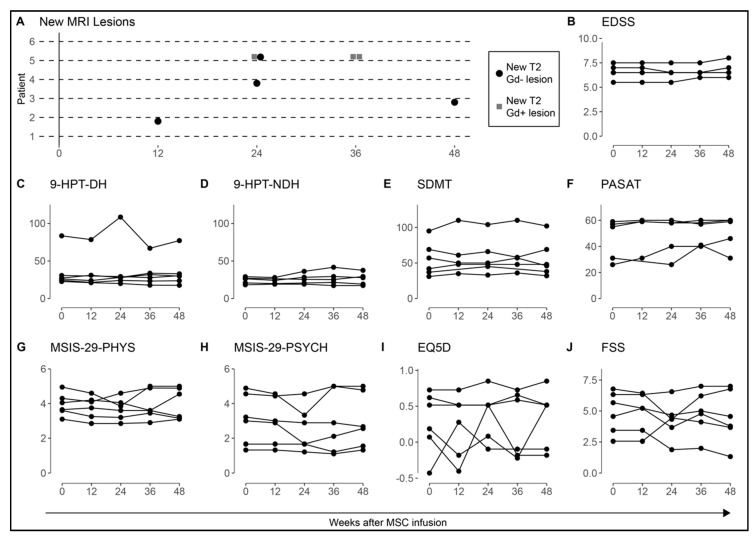
MRI and clinical outcome measures after mesenchymal stromal cells (MSC) therapy. Clinical measures and MRI were performed at baseline, +12, 24, 36, and 48 weeks post MSC infusion in 6/7 MS patients. (**A**) Four patients had a total of seven new T2 MRI lesions in the year following MSC treatment (MRI examination was not performed at +24 and +36 weeks for patient 1). (**B**) The EDSS scores remained without significant change in all patients (*n* = 6) and repeated measures revealed stable hand function (**C**,**D**), cognitive function (**E**,**F**), and quality of life (**G**–**J**) after MSC infusion. The test scores for each time point were compared to the baseline scores using the Wilcoxon Signed Rank test. One patient (patient 7) did not participate in the follow up and was excluded from the analysis. Hand function and cognitive tests were not performed in patient 1 at +12 w. MRI: magnetic resonance imaging, T2: T2 weighted imaging, Gd: gadolinium enhancement. EDSS: expanded disability status scale, 9-HPT-DH: 9 hole peg test–dominant hand, 9-HPT-NDH: 9 hole peg test–non-dominant hand, SDMT: symbol digit modalities test, PASAT: paced auditory serial addition test, MSIS-PHYS: multiple sclerosis impact scale–physiological impact, MSIS-PSYCH: multiple sclerosis impact scale–psychological impact, EQ5D: European quality of life-5 dimensions, FSS: fatigue severity scale.

**Figure 3 jcm-08-02102-f003:**
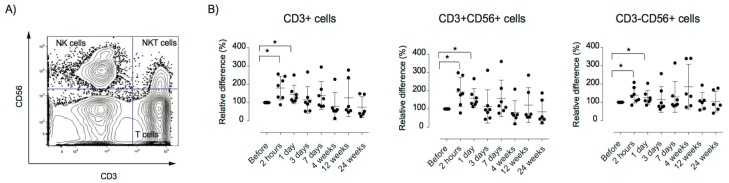
Relative surface expression of CXCR3 after MSC infusion. PBMCs were analyzed using flow cytometry for the expression of the chemokine receptor CXCR3. (**A**) Representative dot plots of the gating strategy. (**B**) Relative expression of mean fluorescence intensity (MFI) of the CXCR3 receptor at the surface of T cells (CD3+CD56−), NKT cells (CD3^+^CD56^+^), and NK cells (CD3−CD56+). Error bars show mean +/− SEM, p-values using the Wilcoxon signed-rank test. The relative distribution of immune cells was calculated as described in the materials and methods sections. PBMC: peripheral blood mononuclear mononuclear cells.

**Figure 4 jcm-08-02102-f004:**
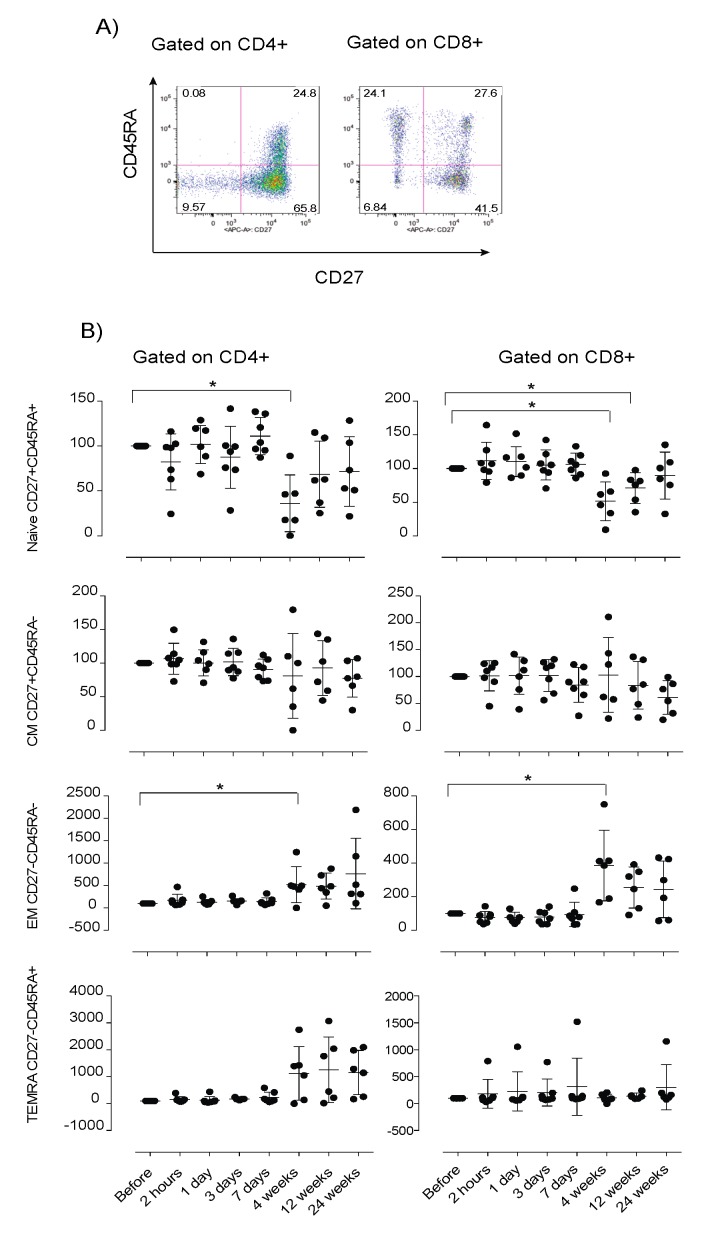
Relative proportions of circulating naïve and memory T cells after MSC therapy. PBMCs were analyzed using flow cytometry for naïve and memory T cells. (**A**) Representative dot plots of the gating strategy of naïve and memory CD4 and CD8 cells, based on the expression of CD27 and CD45RA. CD4 and CD8 cells were gated on CD3+CD56 cells. (**B**) Longitudinal analyses of T cell subsets. Six patients were sampled for 24 weeks, one patient was withdrawn from the study at +4 weeks and was only sampled for 7 days, (+1 day; *n* = 6). Relative distribution of immune cells was calculated as described in the materials and methods sections. Error bars show mean +/− SEM, *p*-values using the Wilcoxon signed-rank test, * *p* < 0.05. PBMC: peripheral mononuclear cells, EM: early memory T cells, CM: central memory T cells, TEMRA: late memory T cells.

**Figure 5 jcm-08-02102-f005:**
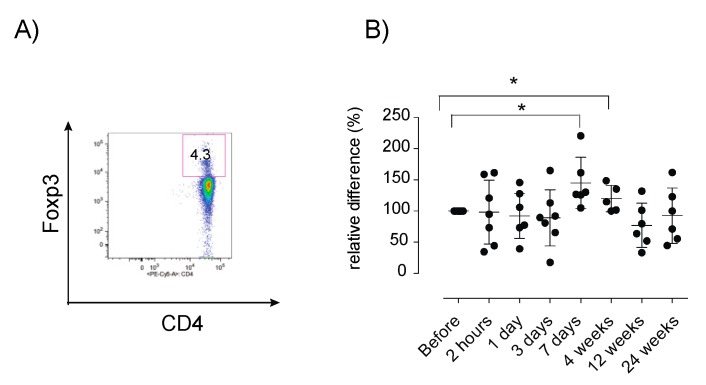
Relative proportion of peripheral FOXP3+ Tregs after MSC therapy. PBMCs were analyzed using flow cytometry for the assessment of the relative proportion of FOXP3+ Tregs in MS patients. (**A**) Representative dot plot show gating strategy of FOXP3+ cells within the CD4+ population. (**B**) Longitudinal analyses of FOXP3+ Tregs. Six patients were sampled for 24 weeks, one patient was withdrawn from the study and was only sampled for 7 days, (+1 day; *n* = 6). Relative distribution of immune cells was calculated as described in the materials and methods sections. Error bars show mean +/− SEM, *p*-values using the Wilcoxon signed-rank test, * *p* < 0.05. PBMC: peripheral blood mononuclear cells, FOXP3 Tregs: Forkhead box P3+ regulatory T cells.

**Figure 6 jcm-08-02102-f006:**
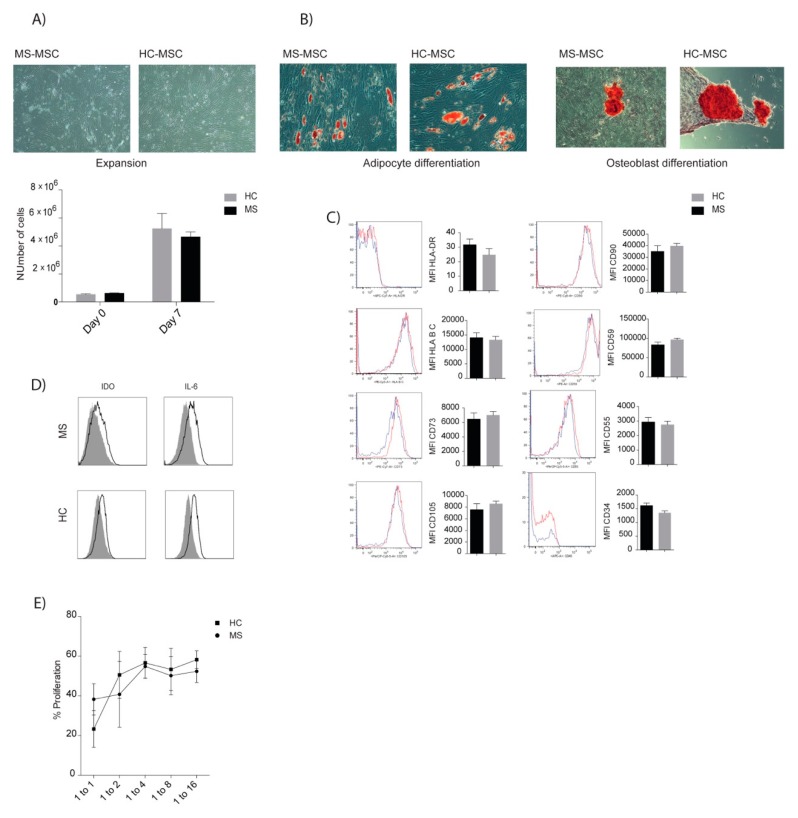
In vitro characterization of BM-MSC of MS patients and healthy donors. (**A**) Cryopreserved MSC from MS and HC (healthy controls) were thawed and expanded for 7 days (MS; *n* = 6, HC; *n* = 5). (**B**) Representative photographs of MSC adipocyte and osteogenic differentiation. (**C**) Cell surface expression of MSC molecules, assessed using flow cytometry, with data expressed as median fluorescence intensity (MFI) ± SD. (**D**). Freshly thawed MSC were stimulated for 72 h with IFN-γ and TNF-α. Intracellular expression of IDO and IL-6 was measured by flow cytometry (black lines). The control grey histogram represents unstained MSC. (**E**) Proliferative capacities of CD3+ cells in the presence or absence of MSC, at ratios of (MSC: CD3^+^) 1:0, 1:1, 1:2, 1:4, 1:8, 1:16 were analyzed after five days of co-culture (*n* = 4/group). Comparisons between MSC donor groups were analyzed using the Mann Whitney test.

**Table 1 jcm-08-02102-t001:** Baseline characteristics of MS study subjects.

Number of Patients	7
MS subdiagnosis, SPMS/PPMS	5/2
Sex, female/male	6/1
Age, median (range)	40 (23–49)
Disease duration, years, median (range)	12 (5–19)
MRI T2 lesion number, median (range)	20 (20–50)
Number of Gd T2, median *	0
Relapses within 2 years, median (range)	1.5 (0–3)
EDSS, median (range)	6.5 (5.5–7.0)
9-HPT dominant hand, median (range)	26.1 (17.7–83.5)
9-HPT non-dominant hand, median (range)	19.6 (18.1–27)
SDMT, median (range)	50 (31–95)
PASAT, median (range)	54 (26–59)
MSIS-29 PHYS, median (range)	3.65 (2.6–4.95)
MSIS-29 PSYCH, median (range)	3 (1.3–4.9)
FSS, median (range)	5.8 (2.6–6.8)
EQ5D, median (range)	0.52 (0.1–0.73)
Prior disease modifying treatment	interferon beta (*n* = 4)
	glatiramer acetate (*n* = 3)
	fingolimod (*n* = 2)
	natalizumab (*n* = 2)
	mitoxantrone (*n* = 1)
	rituximab (*n* = 2)

MS; multiple sclerosis, SPMS; secondary progressive MS, PPMS; primary. progressive MS, MRI: Magnetic Resonance Imaging, Gd; gadolinium enhancing lesions, T2: T2 weighted image, EDSS: expanded disability status scale, 9-HPT-DH: 9 hole peg test–dominant hand, 9-HPT-NDH: 9 hole Peg Test–non-dominant hand, SDMT: symbol digit modalities test, PASAT: paced auditory serial addition test, MSIS-29 PHYS: multiple sclerosis impact scale–physiological impact, MSIS-PSYCH: multiple sclerosis impact scale–psychological impact, FSS: fatigue severity scale, EQ5D: European quality of life-5 dimensions. * Patient 7 excluded.

**Table 2 jcm-08-02102-t002:** Differentially expressed microRNAs after MSC therapy.

+2 h after MSC Infusion			
miRNA	SD	Fold Change	*p*-Value
hsa-miR-193a-5p	1.2	8.1	0.00060
hsa-miR-365a-3p	1.4	4.8	0.0050
hsa-miR-34a-5p	1.1	4.6	0.0020
hsa-miR-100-5p	1.3	4.2	0.0061
hsa-miR-375	1.4	2.5	0.048
hsa-miR-215-5p	0.60	1.8	0.011
hsa-miR-192-5p	0.50	1.6	0.010
hsa-miR-16-5p	0.61	1.5	0.044
hsa-miR-140-3p	0.40	1.3	0.034
hsa-miR-335-5p	1.1	−2.8	0.012
hsa-miR-15b-5p	0.76	−1.8	0.024
hsa-miR-543	0.64	−1.7	0.020
hsa-miR-374b-5p	0.65	−1.6	0.040
hsa-miR-155-5p	0.43	−1.5	0.014
hsa-miR-584-5p	0.35	−1.5	0.0053
hsa-miR-126-3p	0.41	−1.3	0.031
hsa-let-7d-3p	0.28	−1.2	0.031
hsa-miR-30d-5p	0.31	−1.2	0.048
hsa-miR-362-3p	0.33	−1.2	0.049
**+24 h after MSC infusion**			
hsa-miR-332-3p	0.5	−1.4	0.044
**+3 days after MSC infusion**			
hsa-miR-375	2.4	4.9	0.046
hsa-miR-143-3p	0.65	−1.6	0.049
hsa-miR-29a-3p	0.53	−1.5	0.049
hsa-miR-133a-3p	0.21	−1.2	0.021

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
