# Peer review of "Short and Long Term Clinical and Immunologic Follow up after Bone Marrow Mesenchymal Stromal Cell Therapy in Progressive Multiple Sclerosis—A Phase I Study"

_jcm, 2019, doi:10.3390/jcm8122102_

Round 1

Reviewer 1 Report

Iacobaeus et al. describe their findings following their open-label phase I, single-center, pre- and post-comparison study on the safety and clinical response to BM-MSC therapy in MS, using a single injection of autologous cells.   They report that although at 48 weeks follow-up in 5 of 7 patients had new lesions, it seemed that when patients received the treatment in remission, their disease was stable or in one case improved at 24 weeks after treatment. The MSC seemed to be well tolerated and safe but the beneficial effect seemed transient. At certain intervals peripheral blood mononuclear cells were taken from the patients and analysis was performed on immune cell populations.  In addition, cytokine levels and microRNA populations were analyzed. They report an important finding in the one patient who received infusion during a relapse and his/her symptoms worsened.  This observation suggests that selecting the right patients and the right time is critical in MSC therapy. The immune cell population and the microRNA analysis are very important novel additions to the field.

The work is interesting, carefully done and well written. There are a few small comments /questions that might need to be discussed to add to the value of the work.

1. The use of autologous MSCs in an autoimmune disease might not be ideal. What kind of cells were used in the other studies? Are there any data to show if and how autologous and allogeneic cells perform in autoimmune diseases?  If allogeneic cells were found better how would those be chosen?

2. The MSCs were used in passage 1-2 according to the description.  It would be nice to know what this means in population doubling. It should also be discussed, how the same passage could be used for more treatments – if the single treatment will not be enough for long-term effect.  Could then the cells be expanded before the first treatment and then have enough of the same cells for consecutive treatments?

Author Response

Q1.The use of autologous MSCs in an autoimmune disease might not be ideal. What kind of cells were used in the other studies? Are there any data to show if and how autologous and allogeneic cells perform in autoimmune diseases?  If allogeneic cells were found better how would those be chosen?

Answer: We thank the reviewer for this excellent comment. We have added a part into the discussion, in order to further highlight this important topic. Discussion, line 450-458

Q2. The MSCs were used in passage 1-2 according to the description.  It would be nice to know what this means in population doubling. It should also be discussed, how the same passage could be used for more treatments – if the single treatment will not be enough for long-term effect.  Could then the cells be expanded before the first treatment and then have enough of the same cells for consecutive treatments?

Answer: The two questions are both of great interest. Data about population doubling has been added in the Result section 3.2, line 282.

The MSCs harvested from MS patients showed excellent expansion potential in vitro. In the present study, MSCs were harvested at very low passage, since the protocol was written to include only one infusion. However, the number of cells generated could easily be increased to generate 2 or more MSC doses, using the same volume of harvested bone marrow. The discussion has been expanded, line x, with a part about the potentiality to perform repeated MSC infusions in the patients, Discussion, line 545-548.

Reviewer 2 Report

Dear Authors,

The manuscript entitled “Short and long term clinical and immunologic follow up after bone marrow mesenchymal stromal cell therapy in progressive multiple sclerosis – a phase 1 study is a well- designed study that supports the safety and tolerability of IV infused MSCs in MS patients.

However, major revisions must be performed before the study further processed.

Reviewer’s Comments

The majority of the abbreviations have been well addressed in the abstract of the manuscript. The terms should be abbreviated the first time of their use, and then only the abbreviation should be used. Please note the author guidelines regarding the abbreviations and perform the corrections.

A schematic flow chart of the clinical trial must be added to the manuscript , in order to help the readers to better understanding the whole study.

It is not clear described in the manuscript, either in the introduction or materials and methods section if MS patients continued to receive their therapy during the MSC infusion or not. It should be addressed in the section of Materials and Methods – 2.2 MS subjects, healthy MSC donors and ethics.

Materials and Methods – 2.3 MSC therapy, lines 139 – 143. The authors indicated that after thawing, the final MSCs concentration for infusion was 2 x 106 cells / ml. In this part, it should be added that the total number of infused MSCs depends on patient’s body weight.

Materials and Methods – 2.8 In vitro phenotypic and immunosuppressive analysis of MSCs, lines. In my opinion, this part contains both the assays and the results of MSCs’ quality control. This section should be divided in two parts. The first part will include the assays needed for the quality control of MSCs, which belongs to materials and methods section. The second part, which is referring to the results shall be added in the results section.                                                                    In addition, in the introduction section, lines 102 – 104, one of the aims of the study was comparison of the phenotypic and immunosuppressive properties of MS MSCs with HD MSCs. However, the results of this comparison has been provided as supplementary data (Figure S2). It will be better for the readers, these results to be provided in the main manuscript and not as supplementary.

Results – 3.4 Plasma cytokine and miR levels after MSC infusion, lines 293 – 296. The authors presented that altered expression of one miR at 24h post MSC infusion compared to pre-treatment levels. However, it is not described the evaluation of miRs in materials and methods section, and also no results are presented in the results section. So, has it been performed evaluation of differential expression of miRs pre-treatment? Please present these results.

Discussion, lines 379 – 382. The authors described that MSCs after IV infusion, were cleared from circulation and accumulated in the lungs. Did the authors performed such experiments to prove this hypothesis. For example, it is desired to label the MSCs before the IV infusion with superparamagnetic iron oxide (ferumoxides). After the infusion, the trafficking and migration of MSCs could be performed with MRI. For more details, please kindly view the publication of Karussis D, Karageorgiou C, Vaknin-Dembinsky A, Gowda-Kurkalli B, Gomori JM, Kassis I, Bulte JW, Petrou P, Ben-Hur T, Abramsky O, Slavin S. Safety and immunological effects of mesenchymal stem cell transplantation in patients with multiple sclerosis and amyotrophic lateral sclerosis. Arch Neurol. 2010 Oct;67(10):1187-94. doi: 10.1001/archneurol.2010.248.

Yours sincerely

Author Response

Q1. The majority of the abbreviations have been well addressed in the abstract of the manuscript. The terms should be abbreviated the first time of their use, and then only the abbreviation should be used. Please note the author guidelines regarding the abbreviations and perform the corrections.

A1.We have corrected the abbeviations according to the guidelines.

Q2. A schematic flow chart of the clinical trial must be added to the manuscript, in order to help the readers to better understanding the whole study.

A2. Excellent suggestion. A new Figure (Figure 1) has been added to the manuscript with a schematic flow chart of the clinical study to facilitate for the reader to get a better overview of the trial, Result section 3.1, line 261 and (new) Figure 1

Q3. It is not clear described in the manuscript, either in the introduction or materials and methods section if MS patients continued to receive their therapy during the MSC infusion or not. It should be addressed in the section of Materials and Methods – 2.2 MS subjects, healthy MSC donors and ethics.

A3. We thank the reviewer for this important notification and apologize that this important information was not stated clearly. The information was previously only given in the Supplementary data as part of the Inclusion and exclusion criteria. Additional information has been added into Methods section 2.2, line 137-139 and Result section 3.2, line 287-288.

Q4.Materials and Methods – 2.3 MSC therapy, lines 139 – 143. The authors indicated that after thawing, the final MSCs concentration for infusion was 2 x 106 cells / ml. In this part, it should be added that the total number of infused MSCs depends on patient’s body weight.

A4. We thank the reviewer for this suggestion and have added information to clarify that the patients were treated with 1-2x106 cell/kg body weight.

Q5.Method section part 2.3 Line 166. Materials and Methods – 2.8 In vitro phenotypic and immunosuppressive analysis of MSCs, lines. In my opinion, this part contains both the assays and the results of MSCs’ quality control. This section should be divided in two parts. The first part will include the assays needed for the quality control of MSCs, which belongs to materials and methods section. The second part, which is referring to the results shall be added in the results section. In addition, in the introduction section, lines 102 – 104, one of the aims of the study was comparison of the phenotypic and immunosuppressive properties of MS MSCs with HD MSCs. However, the results of this comparison has been provided as supplementary data (Figure S2). It will be better for the readers, these results to be provided in the main manuscript and not as supplementary.

A5. The result section has been changed and the Figure S2 has now been included in the manuscript as Figure 6. Result section 3.5, line 398-407. New Figure 6

Q6. Results – 3.4 Plasma cytokine and miR levels after MSC infusion, lines 293 – 296. The authors presented that altered expression of one miR at 24h post MSC infusion compared to pre-treatment levels. However, it is not described the evaluation of miRs in materials and methods section, and also no results are presented in the results section. So, has it been performed evaluation of differential expression of miRs pre-treatment? Please present these results.

A6. We thank the reviewer for pointing out the lack of clarity. Analyses of miR expression post 24h was performed but showed only 1 differently expressed miR (hsa-miR-332-3p) wich precluded KEGG analyses of overlap between miRs at this time point. We have rectified the previous error and clarified this in the Method section 2.7, line 203 and 210, Result section 3.4, line 347-352 and Table 2 (red text).

Q7. Discussion, lines 379 – 382. The authors described that MSCs after IV infusion, were cleared from circulation and accumulated in the lungs. Did the authors performed such experiments to prove this hypothesis. For example, it is desired to label the MSCs before the IV infusion with superparamagnetic iron oxide (ferumoxides). After the infusion, the trafficking and migration of MSCs could be performed with MRI. For more details, please kindly view the publication of Karussis D, Karageorgiou C, Vaknin-Dembinsky A, Gowda-Kurkalli B, Gomori JM, Kassis I, Bulte JW, Petrou P, Ben-Hur T, Abramsky O, Slavin S. Safety and immunological effects of mesenchymal stem cell transplantation in patients with multiple sclerosis and amyotrophic lateral sclerosis. Arch Neurol. 2010 Oct;67(10):1187-94. doi: 10.1001/archneurol.2010.248.

A8. We thank the reviewer for pointing out that this part of the discussion was unclear. We have unfortunately not performed any experiments to support the hypothesis that MSCs after IV infusion may accumulate in the lung or other tissues. We agree that it would have been of great value if MRI studies had been performed in order to track the fate of IV infused MSCs Such experiments might perhaps have been able to corroborate the findings in the excellent work by Karussis D. et al, in Arch Neurol 2010. We have rewritten the section in the discussion in order to clarify that the hypothesis that MSCs may accumulate in the lungs after IV infusion only relates to previous studies. In addition, a part has been added to discuss the previous detection of MSCs in the CNS of MS patients and EAE mice after IV MSC therapy (described in the publication of Karussis D. et al Arch Neurol 2010) as mentioned by the reviewer), indicating that migration of MSCs into the CNS are associated with their mechanism of action in neuroinflammatory diseases. Discussion, Line 477-487.

Reviewer 3 Report

The manuscript by Iacobaeus et al, details a Phase I MSC transplant study in progressive MS. The n for the study is very small, just 7, with not all participants completing the study. This affects the impact the article will have on the field. It is, however, another MSC trial in the field, which is definitely needed. The addition of the immunological effects following MSC infusion greatly increase the strength of the manuscript and its contribution of the field of literature.

Issues that need to be addressed:

Line 143: Please elaborate on the post-thawing to infusion timing of the MSCs. How long were the MSCs allowed to recover post-thaw prior to infusion? Data from the Galipeau lab suggest that MSCs which are infused shortly after thawing are not fully functional, which may lead to a decrease in their therapeutic impact. His data suggest that a 24-48 hour “recovery” period is best following cell thawing. With this study beginning in 2012, the data from his lab were not yet widely known at the time of protocol development, but the authors need to comment on this potential issue related to cell functioning in terms of clinical results observed during the trial. Line 162: RPMI is not simply “Roswell Park Memorial Institute.” While I have never seen RPMI noted this way in text form, at least add the word “Medium” after Institute to indicate that you are referring to the medium, not the educational Institute, otherwise it is confusing. “RPMI 1640 (Roswell Park Memorial Institute Medium)” or just “RPMI 1640.” Line 265: “…a total of seven new T2 lesions in 5/6 patients over 48 weeks…” From both the description about this excerpt and from Fig 1a, it appears that new T2 lesions were noted in 4/6 participants, not 5/6 (pt 2, pt 3, pt 4, and pt 5). New Gd+ lesions were found in 1/6 (pt 5 only). Please clarify/correct. Line 269: It is a bit of a stretch to say that the EQ5D is “consistent” as it jumps around for a few participants during the study. If the authors mean that the results across all of the PROs were “consistent” across the measures for each participant—that clarification should be made. Figure 1 (c-e): are the data showing impairment (the upper line in these figures) from the same participant? Figure 2b: It appears that there is consistently one or 2 data points above all others—are these from the same individuals or do individuals jump around from high to low? If just one or two individuals—is there anything different about them performance, clinically, or outcome-wise? Lines 360-364 indicate that this is not the person with the infusion mid-relapse—which is what many may assume before reaching the discussion. Line 305: CXCR3 comes out of the blue here without a single statement relative to the importance of this receptor. Add something to say WHY you are looking at this—either here or in the figure legend. Line 306: “…baseline at 48 h post-infusion, (Figure 2).” Figure 2 does not show 48 h post-infusion. The timepoints include 2h post, 24 h post, and 72h post. Please correct the statement to reflect the data shown—or include 48h post data. Line 402: Remove “at” from “apparent until at seven days”

Author Response

Q1 Line 143: Please elaborate on the post-thawing to infusion timing of the MSCs. How long were the MSCs allowed to recover post-thaw prior to infusion? Data from the Galipeau lab suggest that MSCs which are infused shortly after thawing are not fully functional, which may lead to a decrease in their therapeutic impact. His data suggest that a 24-48 hour “recovery” period is best following cell thawing. With this study beginning in 2012, the data from his lab were not yet widely known at the time of protocol development, but the authors need to comment on this potential issue related to cell functioning in terms of clinical results observed during the trial. We thank the reviewer for this important question.

A1. We thank the reviewer for this interesting and important comment. We are aware of the data from the research group of J Galipeau that reported impaired immunosuppressive properties of freshly thawed MSCs. Indeed, the current study begun in 2012 and the study protocol was completed before 2012. The applied procedure for IV MSC infusion was based on our previous experience in clinical trials using cryopreserved MSCs in other diseases (i.e. Graft versus host disease, Diabetes Mellitus). According to the current protocol, the patients received IV infusion of freshly thawed MSCs and no recovery period was applied which may have affected the MSC product quality and therapeutic potency. We have added a part to clarify the MSC infusion procedure in relation to thawing and have added a reference from the Galipeau group (Francois M et al. Cytotherapy 2012, PMID: 22029655). Method section 2.3 line 165 and Discussion line 459-465.

Q2. Line 162: RPMI is not simply “Roswell Park Memorial Institute.” While I have never seen RPMI noted this way in text form, at least add the word “Medium” after Institute to indicate that you are referring to the medium, not the educational Institute, otherwise it is confusing. “RPMI 1640 (Roswell Park Memorial Institute Medium)” or just “RPMI 1640.”

A2. We have corrected the information about the RPMI medium used in the study. Method section 2.6, Line 190. Line 265:

Q3 “…a total of seven new T2 lesions in 5/6 patients over 48 weeks…” From both the description about this excerpt and from Fig 1a, it appears that new T2 lesions were noted in 4/6 participants, not 5/6 (pt 2, pt 3, pt 4, and pt 5). New Gd+ lesions were found in 1/6 (pt 5 only). Please clarify/correct.

A3. We thank the reviewer for notifying this error in the text and apologize for our mistake. As the reviewer points out, the correct number with new T2 lesions are 4/6 participants, not 5/6 and new Gd+ lesions were found in 1/6 patients. Result section 3.3, Line 309-310 Line 269:

Q4.It is a bit of a stretch to say that the EQ5D is “consistent” as it jumps around for a few participants during the study. If the authors mean that the results across all of the PROs were “consistent” across the measures for each participant—that clarification should be made. Figure 1 (c-e): are the data showing impairment (the upper line in these figures) from the same participant?

A4. We agree with the reviewer that the description of the PRO results where unclear and they have now been re-written. Result section 3.3, line 313-315.

Q5.Figure 2b: It appears that there is consistently one or 2 data points above all others—are these from the same individuals or do individuals jump around from high to low? If just one or two individuals—is there anything different about them performance, clinically, or outcome-wise? Lines 360-364 indicate that this is not the person with the infusion mid-relapse—which is what many may assume before reaching the discussion.

A5. An interesting observation. The data points above all others come from the same individuals, the individuals did not jump around from high to low. None of the higher values represented the patient with the infusion mid relapse. There where no difference in clinical performance between the individuals with the higher data points, both patients had PPMS (one female, one male) but their disease history and prior treatments had been different.

Q6.Line 305: CXCR3 comes out of the blue here without a single statement relative to the importance of this receptor. Add something to say WHY you are looking at this—either here or in the figure legend. Line 306: “…baseline at 48 h post-infusion, (Figure 2).”

A6. We fully agree. We were interested in CXCR3 expression due to the previous observation that downregulation of CXCR3 expression on T cells have been suggested to be a mechanism involved in MSC mediated T-cell suppression. This has been clarified in the text. Result section 3.5., line 363-364, and a new reference has been added.

Q7. Figure 2 does not show 48 h post-infusion. The timepoints include 2h post, 24 h post, and 72h post. Please correct the statement to reflect the data shown—or include 48h post data.

A7. We thank the reviewer for notifying our error. The increased expression of CXCR3 was detected at 2h and 24 h and remained to baseline levels at 3 days. Analyses at 48h was not performed. We have rectified the error, Result section 3.5, line 361-362. Line 402:

Q8. Remove “at” from “apparent until at seven days”

A8. The phrase has been corrected, Discussion, line 507.

Round 2

Reviewer 2 Report

Dear Authors,

The majority of the comments have been well addressed.

The whole manuscript has been improved significantly.

Thank you for the performed corrections according to my suggestions.

Author Response

Thank you for your revision of the manuscript.